# High Oxygen Shocking Reduces Postharvest Disease and Maintains Satisfying Quality in Fresh Goji Berries during Cold Storage by Affecting Fungi Community Composition

**DOI:** 10.3390/foods12132548

**Published:** 2023-06-29

**Authors:** Shuangdi Hou, Gaopeng Zhang, Wei Zhao, Jiaxuan Zheng, Min Xue, Yanli Fan, Xia Liu

**Affiliations:** 1State Key Laboratory of Food Nutrition and Safety, Key Laboratory of Food Nutrition and Safety, Ministry of Education of China, College of Food Science and Engineering, Tianjin University of Science & Technology, Tianjin 300457, China; houshuangdi321@163.com (S.H.); zhanggp0501@126.com (G.Z.); fysf_zw@163.com (W.Z.); zhengjia_xuan@126.com (J.Z.); nutxue@163.com (M.X.); 2School of Food & Wine, Ningxia University, Yinchuan 750021, China; fanyanli_fyl@163.com; 3Tianjin Fresh Food and Biological Technology Co., Ltd., Tianjin 300457, China

**Keywords:** fungi community, high oxygen shocking, goji berry, quality

## Abstract

Fresh goji (*Lycium barbarum* L.) berries were treated with high-concentration (50% and 90%) oxygen shocking for 30 min and then stored at 0 ± 0.5 °C for 30 d. Decay, aerobic plate count, firmness, weight loss, total soluble solid (TSS), and titratable acidity (TA) were evaluated during storage. A total of 90% O_2_ shocking more effectively reduced decay and maintained the weight loss and firmness of goji berries. Subsequently, changes in fungi communities were analyzed using high-throughput sequencing (HTS) in the 90% O_2_-shocking and control groups. The results showed that 90% O_2_ shocking retained the richness and diversity of fungi communities and the microbiome was related to the quality properties of the fruit. Thus, we inferred that high oxygen shocking inhibited the development of natural decay and maintained the satisfying quality of goji berries by affecting the fungi community composition, which reduced the growth of pathogenic fungi and harmful saprotrophic fungi in the genera, such as *Filobasidium* sp., *Alternaria* sp., and *Cladosporium* sp. We provide a new insight into the disease development and quality changes during the storage of postharvest goji berries.

## 1. Introduction

Goji (*Lycium barbarum* L.) is a subtropical berry fruit distributed in regions of Asia, Africa, America, and Australia, and is highly popular due to its good nutritional value, antioxidants, and unique flavor [1]. Despite dried goji berry being the primary form consumed for medicine purposes, they have gained recognition as the latest ‘super fruit’ in the worldwide due to their super taste, juicy texture, and more active ingredients compared to the dried products [2]. However, fresh goji berries are vulnerable to fungi contamination, which results in quality deterioration and short shelf life, making them unable to meet the market demand. Many researchers have identified the main pathogenic microorganisms responsible for fresh goji berry postharvest disease, including *Alternaria* sp., *Cladosporium* sp., *Fusarium* sp., and *Fiobasidium* sp. [3,4]. A decrease in the epicuticular wax content of ripe to overripe fruits may promote the exudation of nutrient composition, which stimulate the superficial development of fungi in berries [5]. Thus, microbial infections may relate to softening, water loss, and changes in the nutrient composition of fruits. An effective technology that reduces pathogenic microorganisms in fresh goji berries is urgently needed to decrease the postharvest incidence rate of fresh goji berries and extend their shelf life.

In the last several decades, various methods have been commonly employed to control postharvest diseases, including synthetic fungicides treatment, microbial antagonist agents, natural antimicrobial substances, and physical methods [6,7,8,9,10]. However, the extensive use of chemical and microbial fungicides has aroused issues relating to environmental protection, human health, and pathogen drug resistance. Thus, physical methods are considered the most convenient, safe, feasible, and environmentally friendly approach for preserving berry fruits. Some physical treatments, such as cold shock treatment and heat shock treatment could maintain higher antioxidant capacity and chilling tolerance of vegetables and fruits, such as papaya, tomato, and citrus [11,12,13]. Heat and CO_2_ shock treatments induce changes in the cuticle of peaches [14]. Furthermore, CO_2_ shock treatment reduced the degradation of pectin in the cell wall and decay of strawberries during storage [15]. Numerous studies have shown that continuous high oxygen treatment (higher than 60%) significantly suppresses microbial growth and delays the decay of berries [10,16]. Short-term 80% O_2_ treatment maintains the quality of fresh-cut potatoes by suppressing the activity of polyphenol oxidase (PPO) and enhancing antioxidant capacity [17]. The combined application of 80–90 kPa O_2_ together with either 10 or 20 kPa CO_2_ has an inhibitory effect on the growth of microorganisms, such as *Botrytis cinerea*, *Escherichia coli*, *Salmonella enteritidis*, *Pseudomonas fluorescens*, and *S. enteritidis* [18,19]. 

Altering the microbial diversity on the surface of fruits can effectively reduce the risk of postharvest deterioration and pathogenic contamination [4,9]. The microbiome of fruits are affected by many other factors, such as climate, cultivation, single fruit differences, and treatment methods [20]. For example, antagonistic yeast *Pichia kudriavzevii* delayed decay in cherry tomatoes by changing the fungi community succession of predominant pathogens [9]. The microbial community on grape epidermis was affected by the fruit development stage of the grapes as well as the location weather of the vineyard [21]. Additionally, hot air treatment maintains Chinese bayberries’ quality by affecting fungi community composition [6]. The rapid developments of fruits’ microbiomes have dramatically deepened our understanding about postharvest biology. However, at present, no studies have focused on the effects of high oxygen shocking on fruit quality control and the postharvest fungi communities of the goji berry fruit during storage. 

In this study, the effects of high oxygen shocking on decay, aerobic plate count, firmness, weight loss, total soluble solids (TSS), and titratable acidity (TA) were analyzed in fresh goji berries during storage at 0 ± 0.5 °C. This work aimed to investigate the effects of high oxygen shocking on the fruit quality and fungi community composition and dynamics. To address a piece of knowledge about the goji berries epidermis microbiota, an internal transcribed spacer region (ITS) was used to identify fungi species.

## 2. Materials and Methods

### 2.1. Materials and Postharvest Treatment

Goji berry fruits (*Lycium barbarum* L. ‘Ningqi No. 7’) were sourced from a local orchard located in Zhongning, Ningxia, China. Once the fruits were packed into plastic boxes, they were immediately delivered to the laboratory of Tianjin University of Science and Technology. Then, they were precooled at 0 ± 0.5 °C for 24 h.

Fruits selected based on size, color, and lack of physical damage were randomly divided into two groups. The high-oxygen-shocking-treated fruit was placed in sealed tanks, which first vented the original air with N_2_, and then 50% and 90% oxygen concentrations were injected into the tanks for 30 min. The concentration of the atmosphere was tested using an O_2_/CO_2_ analyzer (PBI-Dansensor A/S, Danish, Ringsted, Denmark). The control fruit was dipped in the air condition. After treatment, the fruits in the three groups were stored at 0 ± 0.5 °C, and the characteristics were assessed every 6 days. Three replicates were prepared for samples.

### 2.2. Aerobic Plate Count

The aerobic plate count was conducted following the GB 4789.2 standard operation procedure of China (http://down.foodmate.net/standard/yulan.php?itemid=50367) (accessed on 5 July 2021). Weigh 10 g sample in 90 mL sterilized buffer liquid, homogenized. Aliquots of 200 μL of decimal dilutions of the sample are mixed with culture medium in Petri dishes. Test three parallel plates for each dilution. After incubation at 36 °C for 48 h, the colonies are counted and the number of microorganisms per gram of the sample is calculated. Calculate the amount as the weighted mean from the successive dilutions containing between 30 and 300 colony-forming units (CFU). For plates below 30 CFU record the specific number of colonies; plates greater than 300 CFU cannot be recorded. The results were expressed as the negative logarithm of the CFU of samples. The aerobic plate count was evaluated via Equation (1).
(1)$$N=\frac{\sum C\;}{\;\left(N1+0.1N2\right)\;d}$$
where N—the CFU of samples; ∑C —the total CFU of plates; N1—the number of low dilution plates; N2—the number of high dilution plates; and d—dilution factor (low dilution plates).

### 2.3. Determination of Decay and Weight Loss

Fruit decay and weight loss were determined following [22]. Fruit decay incidence was calculated as the ratio of the decayed fruit number to the total number of fruit. Any goji berry fruits with visible mold growth were considered decayed. Weight loss was evaluated via Equation (2).
(2)$$\mathrm{Weight}\;\mathrm{loss}\;\%=\frac{\left(a\;-\;b\right)\times 100\;}{\;a}$$
where a is the initial weight and b is the final weight.

### 2.4. Determinations of Firmness

The firmness of fresh goji berries was determined using the GY-4 fruit durometer (Toppu Instrument Co., Ltd., Hangzhou, China). A cylindrical aluminum probe (P/11) with a diameter of 11 mm was used to measure the firmness of the fruit [23]. The firmness was expressed by the peak force, denoted as Newtons (N).

### 2.5. Determinations of Total Soluble Solids (TSS)

The TSS was measured using a Pocket refractometer (PAL-1, ATAGO, Tokyo, Japan) [24]. After the fruit was juiced through a manual juicer it was placed on the Pocket refractometer to read and record the TSS value.

### 2.6. Determinations of Titratable Acidity (TA)

TA was determined according to [25]. A total of 10 g of goji fruit was extracted with 50 mL distilled water and kept for 30 min, and then the filtered solution (rapid qualitative filter paper, diameter 20–25 μm) was obtained. Standardized sodium hydroxide (NaOH, 0.1 mol L^−1^, pH 8.1) was used to titrate the filtrate containing two drops of phenolphthalein solution until the pink color lasted for 30 s. TA was presented as a percentage of malic acid and the convert coefficient was 0.067 g/mmol.

### 2.7. Fungus Community Dynamics

#### 2.7.1. Collection of Surface Fungi Communities

The total fungi communities of goji berries was collected at 0, 6, 12, 18, 24, and 30 d storage at 0 ± 0.5 °C. The communities were obtained using a filter funnel with a microporous membrane (diameter 50 mm, pore diameter 0.45 μm) to filter the washing water of the goji berries. Each treatment combination included three replications, each consisting of 30 individual fresh goji berries. Vacuum filtration was used to concentrate microbial communities onto the microporous membrane, which was then stored at −80 °C.

#### 2.7.2. DNA Extraction, PCR Amplification, and ITS Sequencing

Total genome DNA from samples was extracted using the CTAB (Cetyl-trimethyl ammonium bromide) method [26]. ITS1 genes of distinct regions were amplified using specific primer (ITS-1737 F: 5′-GGAAGTAAAAGTCGTAACAAGG-3′, ITS-2043 R: 5′-GCTGCGTTCTTCATCGATGC-3′) with the barcode [9]. Forward and reverse primers (0.2 μM), and template DNA (10 ng) were used for PCR reactions carried out using the Phusion^®^ High-Fidelity PCR Master Mix (New England Biolabs, Ipswich, MA, USA). Thermal cycling consisted of initial denaturation at 98 °C for 1 min, followed by 30 cycles of denaturation at 98 °C for 10 s, annealing at 50 °C for 30 s, elongation at 72 °C for 30 s, and finally 72 °C for 5 min. PCR products were mixed at equidensity ratios for purification using the Qiagen Gel Extraction Kit (Qiagen, Munich, Germany). Sequencing libraries were generated using TruSeq^®^ DNA PCR-Free Sample Preparation Kit (Illumina, Hayward, CA, USA) following the manufacturer’s recommendations and index codes were added. The library quality was assessed on the Qubit @ 2.0 Fluorometer (Thermo Scientific, Shanghai, China) and Agilent Bioanalyzer 2100 system. Finally, the library was sequenced on an Illumina NovaSeq platform and 250 bp paired-end reads were generated.

#### 2.7.3. Bioinformatic Analysis

Paired-end reads, merged using FLASH (VI.2.7, http://ccb.jhu.edu/software/FLASH/) (accessed on 10 August 2022) [27], were allocated to samples based on their unique barcode and truncated by cutting off the barcode and primer sequence. Quality filtering on the raw tags was performed under specific filtering conditions to obtain the high-quality clean tags according to the QIIME (V 1.9.1, http://qiime.org/scripts/split_libraries_fastq.html) (accessed on 10 August 2022) quality-controlled process [28,29]. The tags were compared with the reference database (Silva database, http://www.arb-silva.de/) (accessed on 10 August 2022) using the UCHIME algorithm (http://www.drive5.com/usearch/manual/uchime_algo.html) (accessed on 12 August 2022) to detect chimera sequences, and then the chimera sequences were removed [30,31]. Then, the effective tags were finally obtained.

Uparse software (Uparse v7.0.1001, http://drive5.com/uparse/) was used for sequence analyses [32]. Sequences with ≥97% similarity were assigned to the same Operational Taxonomic Units (OUTs). With regard to each representative sequence, taxonomic information was annotated through the Silva Database (http://www.arb-silva.de/) (accessed on 13 August 2022) [33] based on the Mothur algorithm. All alpha diversity indices of samples were calculated using QIIME (Version 1.9.1) and displayed using R software (Version 2.15.3). Beta diversity on weighted UniFrac was calculated using QIIME software (Version 1.9.1) [34]. Additionally, Nonmetric Multidimensional Scaling (NMDS) is a nonlinear model based on Bray–Curtis distance. The Bray–Curtis distance was used to calculate the difference distance between samples [35]. Principal coordinates analysis (PCoA) was used to reduce the dimension of multidimensional microbial data and show the trend of changes through the distribution of samples on the coordinate axis [36]. NMDS and PCoA were plotted using R software (Version 2.15.3). Linear discriminant analysis (LDA) effect size (LEfSe) analysis used LEfSe software (Version 1.0). The threshold for feature discrimination was a logarithmic LDA score of 4.0 [37]. Analysis of the Pearson correlation coefficient (PCC) was performed on selected variables of high oxygen shocking and control treatments. The correlation coefficient was visualized in −1 to +1. *p* value ≤ 0.05 or *p* Value ≤ 0.01 indicated the data was significant [38].

### 2.8. Statistical Analysis

Data were analyzed through a two-way ANOVA analysis of variance. SPSS 22.0 was used for Duncan’s multiple range tests to compare the mean separations. The data were expressed as mean ± SD, and *p* ≤ 0.05 was considered to illustrate statistical significance. All experiments were carried out in triplicates. 

## 3. Results

### 3.1. High Oxygen Shocking Maintained the Good Appearance Quality of Goji Berries

For the overall visual of fruits’ changes, mold was observed in the control group and 50% O_2_-shocking fruits after 12 d, whereas this was less evident in the 90% O_2_-shocking group (Figure 1A). Aerobic plate count in 50% and 90% O_2_-shocking fruits was also significantly lower than the control after 12 d (Figure 1B) (*p* ≤ 0.05). A maximum value of 4.2 ± 0.01 log10 CFU/g, 5.2 ± 0.01 log10 CFU/g, and 6.5 ± 0.01 log10 CFU/g were observed at 30 d in 90% O_2_ shocking, 50% O_2_ shocking, and the control group, respectively. Meanwhile, after 12 days of storage, 50% and 90% O_2_-shocking-treated fruits had significantly less decay than the control (*p* ≤ 0.05). Notably, decay on the 24th day was 4.65%, 11.37%, and 14.03% in 90% O_2_ shocking, 50% O_2_ shocking, and the control group, respectively. These results showed that high oxygen shocking significantly inhibited microorganisms and decay development in goji berries during cold storage, and 90% O_2_ shocking had the best effect on preservation. 

### 3.2. High Oxygen Shocking Maintained the Good Physiological Quality of Goji Berries

The firmness of goji berries decreased gradually in the entire storage time (Figure 2A). High-O_2_-shocking samples significantly maintained a higher firmness than the control (*p* ≤ 0.05), especially in the 90% O_2_-shocking group. Compared with the control group, weight loss was reduced 0.15% and 0.27% by 50% and 90% O_2_ shocking during the 30 days of storage, respectively (Figure 2B). For TSS and TA, the contents showed a similar wave shape trends in the three groups (Figure 2C,D). Compared to the control, a relatively lower content of TSS and a higher content of TA were shown in high-O_2_-shocking samples at the initial stage. Afterward, 50% O_2_ shocking enhanced the accumulation of TSS before 18 d but decreased sharply at 30 d. While the TSS and TA contents had no significant difference between the 90% O_2_-shocking and the control groups. It indicated that 90% O_2_ shocking mainly has a positive effect in initial storage on TSS and TA, which stimulated the key metabolic pathways and delayed the maturity and senescence of goji berries. Therefore, the treatment with 90% O_2_ shocking was adopted to explore the fungi community composition of goji berries based on their better physiological properties.

### 3.3. High Oxygen Shocking Altered the Fungi Community Succession of Goji Berries’ Surface

In this study, high-throughput sequencing (HTS) was used to analyze the effect of high oxygen shocking on the fungi community in goji berry epidermises, which is widely used for the analysis of microbial communities in postharvest fruit. All of the effective tags were clustered for OTUs with 97% similarity, and then species annotation was performed for the sequences of OTUs. A total of 1123 OTUs were produced. The fungi rarefaction curve, rank abundance, and species accumulation boxplot tended to be flat (Appendix A), demonstrating that deep sequencing provided good overall OTU coverage and the sequence number was abundant. The common and unique OTUs of control and high-oxygen-shocking groups were shown in Figure 3A. The initial sample was analyzed with 581 OTUs (0 d). The unique OTUs of the high-oxygen-shocking group were 174, 154, 241, 192, and 150 at 6 d, 12 d, 18 d, 24 d, and 30 d, respectively (Figure 3A). Additionally, there were 143, 142, 63, 80, and 51 unique OTUs in the control samples at 6 d, 12 d, 18 d, 24 d, and 30 d, respectively (Figure 3A). The individual dispersion was higher in the high-oxygen-shocking group than the control nearly throughout the storage duration, and the high oxygen shocking effectively increased the differences between the two groups. Especially after 18 days, there was a significant difference in species diversity found between the two groups, which was due to the initiation of a shift in community succession in response to the high oxygen shock. During the entire storage period, the specific OTUs of the high-oxygen-shocking group increased first at 18 d and then decreased; however, it sharply decreased first at 18 d, then increased and finally decreased in the control. This phenomenon declared the different processes of microbial ecology community species structure changes during storage by high oxygen shocking disturbance. The fungi community structure had more common OTUs in the high-oxygen-shocking group (180 OTUs) than the control (149 OTUs) (Figure 3B,C) due to the disruption of outbreak of preponderant organisms by high oxygen shocking.

### 3.4. High Oxygen Shocking Caused Shifts in Fungi Abundance and Diversity of Goji Berries’ Surface

Alpha diversity analysis indexes (Shannon, Simpson, Chao1, ACE, goods_coverage, and PD_whole_tree) were measured to analyze the abundance and diversity of the microbial community in goji berries. The same wave shape trend of Observed_otus, Shannon, Simpson, Chao1, ACE, and PD_whole_tree indices of the control and high-oxygen-shocking groups was shown in Table 1. Except at 0 d, the fungi communities in the samples tended to decrease in alpha diversity during storage. Observed_otus, ACE, and chao1 indexes of high-oxygen-shocking samples were higher than the control, indicating that high oxygen shocking maintained the richness of the community. The Shannon, Simpson, and PD_whole_tree indices for the high-oxygen-shocking samples were also higher than the control, meaning that high oxygen shocking maintained community diversity and the uniformity of species distribution. Moreover, goods_coverage was >99% in all samples, demonstrating that the sequencing coverage was high. As shown, the diversity of fungi communities was significantly altered by high oxygen shocking.

### 3.5. High Oxygen Shocking Altered the Fungi Community Structure of Goji Berries’ Surface

As shown in Figure 4A, the species diversity of two group samples was compared using the weighted UniFrac (upper) and unweighted UniFrac coefficients (lower) of a heat map representing the beta diversity index. Except for 0 d, the weighting algorithm of difference coefficients between the high-oxygen-shocking and control group were 0.438, 0.318, 0.494, 0.487, and 0.464 at 6 d, 12 d, 18 d, 24 d, and 30 d, respectively. These values indicate that the difference in species diversity between the two groups sharply decreased first at 12 d, then increased at 18 d, and gradually decreased towards the end storage period (Figure 4A). The results showed that the high oxygen shocking exerted an effect on the composition of microorganisms on the surface of goji berries, particularly at 18 d.

NMDS is a nonlinear model based on Bray-Curtis distance [39]. In this dataset, the gradual increase in the MDS1 values along level coordinates with storage time was shown in the control and high-oxygen-shocking fruit (Figure 4B). Notably, the high-oxygen-shocking group was expressed by lower MDS1 values and a relatively slower rate of rise. Additionally, PCoA, based on the weighted UniFrac distance matrix at the OTU level, was used to assess the sample’s fungi community similarity [40]. As shown in Figure 4C, the proportion of the first principal component is 50.67% and the second component is 16.56%. An analysis of similarities showed that control and high-oxygen-shocking fruit were significantly different in fungi community structure (*p* ≤ 0.05), and different goji berry fruit samples were strongly separated. It indicated that high oxygen shocking dramatically affected the microbial community composition on the surface of goji berries.

### 3.6. High Oxygen Shocking Changed the Species Distribution of Goji Berries’ Microbiota

The core microorganism of the goji berries epidermis identified according to the abundance-occupancy distribution, which was composed of the most important taxa (Figure 5). A total of 939 OTUs (83.62%) were annotated to the kingdom level and 184 OTUs (16.38%) were annotated as unclassified. The proportion of phylum, class, order, family, genus, and species levels were all 780 OTUs (69.46%). At the fungi phylum level, *Ascomycota*, *Basidiomycota*, and *Glomeromycota* were the dominant phyla (Figure 5A). Meanwhile, *Dothideomecetes*, *Tremellomycetes*, and *Agaricomycetes* were the dominant classes; *Filobasidiales*, *Pleosporales*, and *Phallales* were the dominant orders; *Filobasidiaceae*, *Pleosporaceae*, and *Phallaceae* were the dominant families; and *Filobasidium magnum*, *Alternaria alternata*, and *Lysurus cruciatus* were the dominant species. At the genus level, the top 30 microbial taxa and relative abundance were shown in Figure 5B; *Filobasidium* sp., *Alternaria* sp., and *Lysurus* sp. were the dominant genera.

Throughout the storage duration, *Filobasidium* sp. (24.43%), *Alternaria* sp. (28.57%), and *Aureobasidium* sp. (10.94%) were the most predominant genera in the control fruit (Figure 5B and Appendix A). The relative abundance of *Filobasidium* sp. was increased from 6.57% at 0 d to 43.81% at 24 d and decreased at 30 d (36.00%). *Aureobasidium* sp. was increased from 5.61% at 0 d increased to 17.65% at 18 d and decreased at 30 d. *Alternaria* sp. was decreased before 18 d (from 39.11% decreased to 22.58%) and maintained a stable level in anaphase storage.

In high-oxygen-shocking fruit, the relative abundance of *Filobasidium* sp. (17.52%) was lower than the control for most of the storage time. The dominant genus accounts for 35.98% and 20.85% in the control and high-oxygen-shocking group at 30 d, respectively (Appendix A). Meanwhile, *Alternaria* sp. was significantly inhibited at an early stage by high oxygen shocking, and tended to be steady in anaphase storage. The relative abundance of *Aureobasidium* sp. was sharply enriched before 12 d and decreased by the end of storage. 

For other species, such as *Udenimyces* sp., *Markia* sp., *Cladosporium* sp., *Didymella* sp., and *Vishniacozyma* sp., there were significant differences between the two groups. Among them, *Markia* sp., *Cladosporium* sp., and *Vishniacozyma* sp. were enriched in the control fruit, while *Udenimyces* sp. and *Didymella* sp. were predominant in high-oxygen-shocking fruit. Additionally, *Lysurus* sp. was the most abundant genera initially (21.81% at 0 d) and their abundance sharply decreased after 6 d (<0.06% at 30 d) in both groups, which may have occurred because cold temperatures are not conducive to its survive.

To find the significantly different species in abundance (i.e., biomarkers) between the high-oxygen-shocking and control fruit, LEfSe was performed on microorganisms from the samples. A total of 48 fungi taxa were shown by LEfSe with significant differences. As shown in Figure 5C, there were five biomarkers (*Lysurus cruciatus*, *Agaricomycetes*, *Phallales*, *Phallaceae*, and *Lysurus* sp.) enriched in the samples at 0 d (Figure 5(Ca)). A total of 25 and 17 biomarkers were remarkably enriched in the control group and high-oxygen-shocking group, respectively. Except at 0 d, *Didymella* sp. (6 d), *Aureobasidium* sp. (12 d), *Filobasidium* sp. (24 d), and *Udeniomyces* sp. (30 d) were significantly enriched in high-oxygen-shocking fruit at the genus level (Figure 5(Ca)). For the control group, *Didymella* sp. and *Alternaria* sp. (6 d), *Cladosporium* sp. (12 d), *Vishniacozyma* sp. *and Filobasidium* sp. (24 d), and *Mrakia* sp. (30 d) were remarkably enriched at the genus level (Figure 5(Cb)). Additionally, the highest number of significant genera was observed at 24 d, including 10 biomarkers in the control. These results were consistent with the analysis of species distribution.

### 3.7. Correlation Analysis between Microorganisms with Quality Properties

The microbial content of the epidermis directly affects the quality of goji berries during storage. Analysis of the PCC was used to investigate the correlation of dominant genera (genus level, top 10) with the quality properties of goji berries [41]. As shown in Figure 6, some fungi genera demonstrate associations with decay, aerobic plate count, firmness, weight loss, TSS, and TA. Among them, *Filobasidium* sp., *Mrakia* sp., and *Vishniacozyma* sp. exhibit positive correlations with decay, aerobic plate count, and weight loss, while displaying negative correlations with firmness. Conversely, *Alternaria* sp. and *Cladosporium* sp. exhibit negative correlations with decay, aerobic bacterial count, and weight loss, while showing positive correlations with firmness. Furthermore, *Lysurus* sp. demonstrates a positive correlation with firmness and a negative correlation with aerobic plate count and weight loss. Notably, there were strong corrections between dominant genera; *Filobasidium* sp. was positively correlated with *Vishniacozyma* sp. (*p* ≤ 0.05), *Alternaria* sp. and *Cladosporium* sp. (*p* ≤ 0.01), and *Didymella* sp. (*p* ≤ 0.05). *Alternaria* sp. was positively associated with *Lysurus* sp. (*p* ≤ 0.05) and *Cladosporium* sp. (*p* ≤ 0.01) and negatively associated with *Vishniacozyma* sp. (*p* ≤ 0.05), which have an important effect on the function and ecological role of the community.

## 4. Discussions

Postharvest fungi contamination causes substantial losses of fruits and vegetables during storage. In recent years, technological advancements have enabled the direct examination of the fruits’ microbiome, allowing for insights into how postharvest treatments mitigate disease and decay [6]. In this study, high oxygen shocking effectively inhibited the aerobic plate count and decay. Three principal inhibitory mechanisms models were supposed: the first model was 90% O_2_ shocking (30 min) and suppressed the growth of low-oxygen consumption bacteria or anaerobic bacteria; the second model was 90% O_2_ shocking (30 min) and enhanced the resistance of fruits, hindered the contamination of pathogens, inhibited the pathogens absorption of fruit nutrients, and, thus, reduced the aerobic plate count and decay; the third model disturbed the changes in microbial ecological community species structure during storage, and inhibited the growth of pathogenic fungi. 

Here, high oxygen shocking improving the good quality properties of fresh goji berries, such as firmness, weight loss, and TA. Some studies have reported that the cell wall is the first barrier of pathogen infection. However, it would be solubilized by the cell-wall-degrading enzymes during storage and lead to decreased firmness, disease the resistance, and increase the decay incidence of fruits [42,43]. Thus, higher firmness can help to delay fruit softening and improve fruit resistance. Moreover, modification of storage atmospheres has the ability to induce plant defensive responses and enhance disease resistance in postharvest commodities [44], which support the second model. 

From the perspective of microbial metagenomic analysis, 939 OTUs in the samples were identified via HTS, indicating that the quality properties of the fruits are associated with these fungi species, which was further proven by Pearson correlation analysis (Figure 6). Notably, compared to the control, more healthy fruit appeared in the high-oxygen-shocking groups (Figure 1 and Figure 2). The microbial alpha diversity of goji berry epidermises was consistent with the results of previous studies [9,45]. Meanwhile, previous reports have indicated a correlation between microbiome diversity and plant health. Diseased plants tend to exhibit a lower diversity of microbes compared to healthy plants, which is associated with the activation of plant immune signals [46]. Additionally, the beta diversity (heatmap, PCoA, and NMDS) and the analysis of LEfSe also exhibited the diversity and community structure and differences in abundance between 90% O_2_-shocking and control groups. These data illustrated that the composition of the fungi microbial communities in the goji berry epidermis formed unique patterns with different storage times as a result of high oxygen shocking.

LEfSe and Pearson correlation analyses indicated that *Vishniacozyma* sp., *Filobasidium* sp., and *Mrakia* sp. were enriched in control fruit, which may accelerate the decay and weight loss of goji berries (Figure 5C and Figure 6). Among them, *Filobasidium* sp. was reported as pathogenic fungi in some plants [47]. Pathogenic fungi pose a threat to plant health and result in major production and economic losses in agriculture [48,49]. Additionally, some saprophytic fungi were inhibited by high oxygen shocking, especially before 12 d, such as *Alternaria* sp. and *Cladosporium* sp. Both of them are responsible for economic losses in numerous crops, causing postharvest rots and other disease symptoms during storage, such as Alternaria rot and Cladosporium rot of fruits [50,51]. High oxygen shocking significantly enriched the presence of *Aureobasidium* sp. in the fruit during early storage. *Aureobasidium* sp., with effective antagonistic properties against postharvest fruit pathogens, have been used to control postharvest diseases in citrus, tomatoes, and apples [52,53,54]. Presumably, it antagonized fruit postharvest pathogens, such as *Penicillium* spp., *Botrytis cinerea*, and *Colletotrichum acutatum* [52,55,56,57]. Based on our analysis of fungi communities, we inferred that high oxygen shocking inhibits pathogenic fungi and harmful saprotrophic fungi, which conduced less disease and decay and maintained the well quality of goji berries. These results confirmed that high oxygen shocking has significant potential for maintaining the postharvest quality of fresh goji berries, as an effective, safe, low-cost, convenient, and environmentally friendly method for berry preservation. Our findings hold great promise for the advancement of postharvest fruit preservation technology. In addition, unidentified fungi genera (Appendix A) should not be overlooked, and further studies are required to gain a comprehensive understanding of their role.

## 5. Conclusions

Our research first explored high oxygen shocking on reducing the occurrence of disease and decay and maintaining the quality properties of goji berries during cold storage by affecting autoresistance and fungi community composition. In general, high oxygen shocking effectively reduced decay, maintained weight loss and firmness, and delayed the maturity of goji berries during storage. Additionally, it caused shifts in the microbial diversity and composition of goji berry epidermises. *Filobasidium* sp., *Alternaria* sp., *Lysurus* sp., *Aureobasidium* sp., *Udeniomyces* sp., *Mrakia* sp., *Cladosporium* sp., *Didymella* sp., and *Vishniacozyma* sp. were the main widespread genera in most goji berry samples. Concurrently, high oxygen shocking inhibited the pathogenic fungi and harmful saprotrophic fungi by promoting the growth of antagonistic postharvest pathogens. The changes in fungi community succession inhibited the development of pathogenesis in goji berries. Based on these results, this study provides insights into the mechanism by which high oxygen shocking controls postharvest losses caused by fungi in fresh fruit.

## Figures and Tables

**Figure 1 foods-12-02548-f001:**
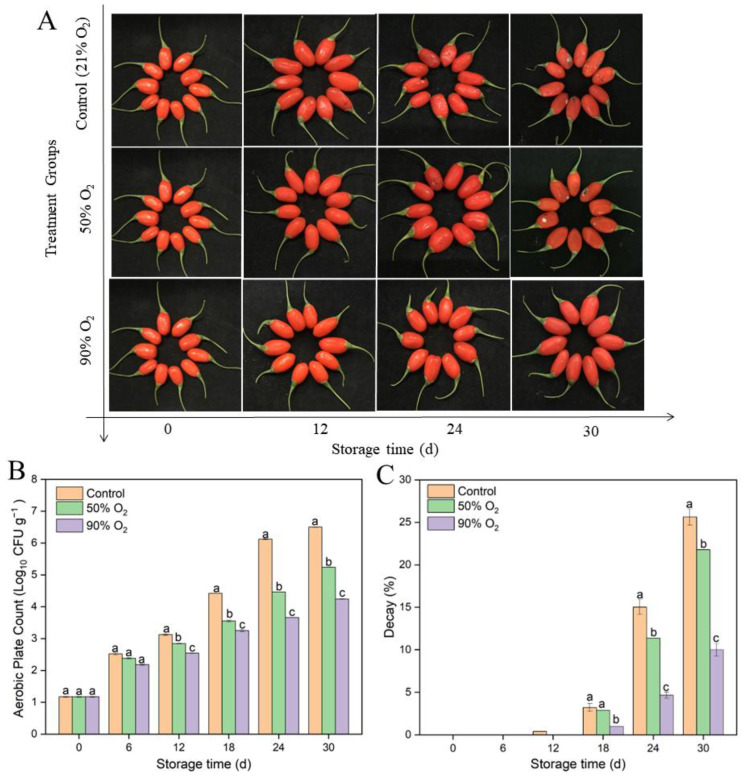
Effects of high oxygen shocking on overall visual quality (**A**), aerobic plate count (**B**), and decay (**C**) of goji berries during storage. Values are means ± SD. Values followed by different letters in the same column are significant (*p* ≤ 0.05).

**Figure 2 foods-12-02548-f002:**
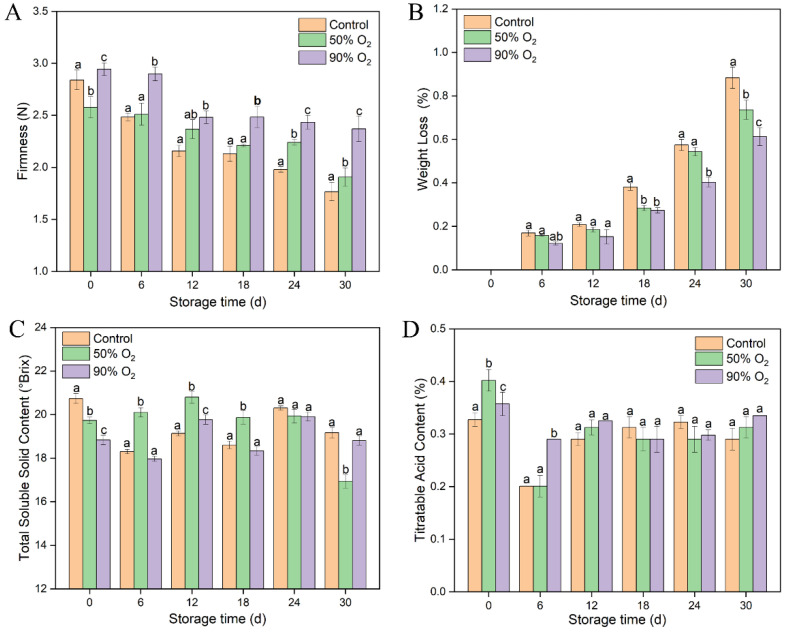
Effects of high oxygen shocking on firmness (**A**), weight loss (**B**), TSS (**C**), and TA (**D**) of goji berries during storage. Values followed by different letters in the same column are significant (*p* ≤ 0.05).

**Figure 3 foods-12-02548-f003:**
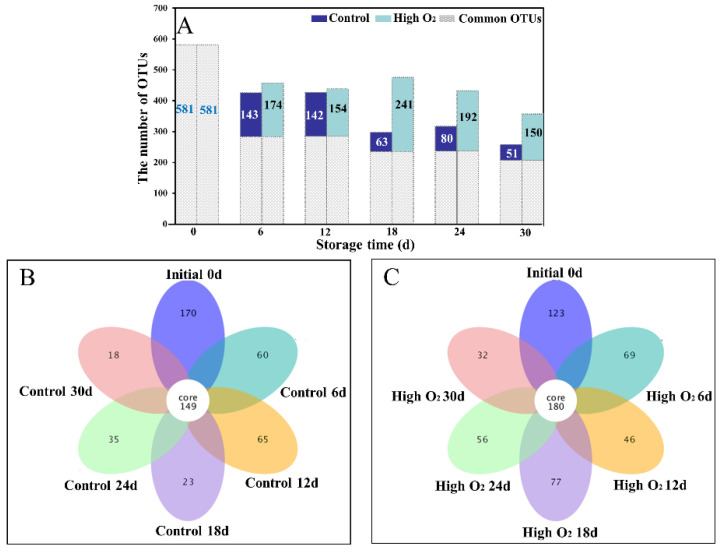
The common and specific fungi OTUs on the surface of goji berries at 0, 6, 12, 18, 24, and 30 d storage (**A**). The flower diagrams are based on OTUs with control and high-oxygen-shocking-treated fruit in (**B**,**C**), respectively.

**Figure 4 foods-12-02548-f004:**
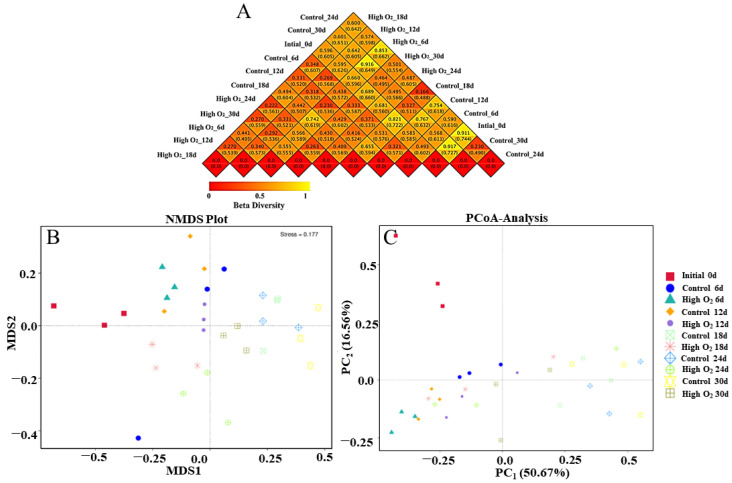
The heat map of beta diversity index (**A**), NMDS analysis (**B**), and PCoA analysis (**C**).

**Figure 5 foods-12-02548-f005:**
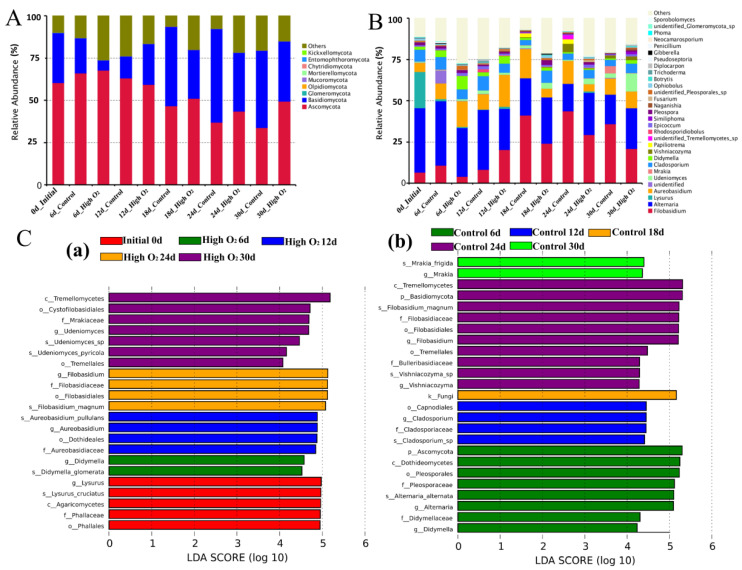
Relative abundance (%) of the predominant fungi at the phylum (**A**) and genus (**B**) levels; LEfSe analysis: k, kingdom; c, class; o, order; f, family; g, genus (**C**), (**a**) represents initial samples and high oxygen shocking samples, (**b**) represents control samples.

**Figure 6 foods-12-02548-f006:**
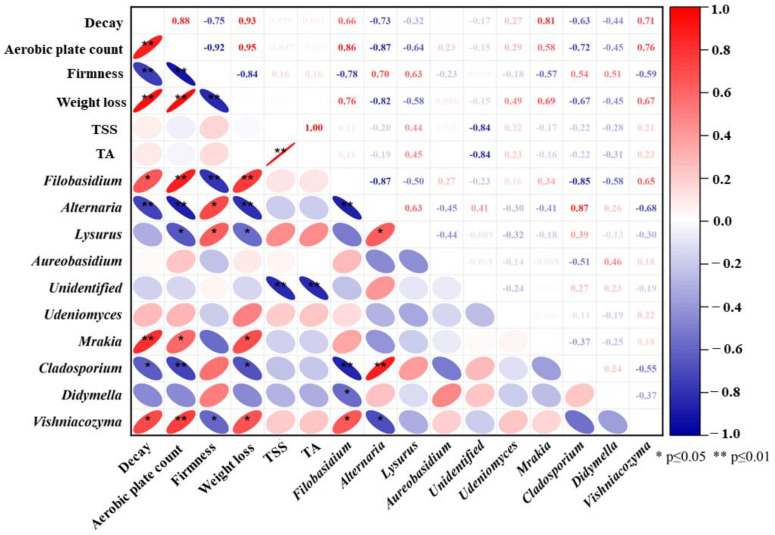
Analysis of Pearson correlation coefficient for quality properties with the dominant genera (top 10) of goji berries at 0 ± 0.5 °C storage. Red indicates positive correlation, blue indicates negative correlation, and white indicates nonsignificant correlation.

**Table 1 foods-12-02548-t001:** Alpha-diversity index of fungi communities associated with the surface of goji berry fruit during 0 ± 0.5 °C storage.

Samples	Observed Otus	Shannon	Simpson	Chao1	ACE	Goods Coverage	PD Whole Tree
Initial_0d	360.00 ± 10.54 ^d^	3.23 ± 0.07 ^abcd^	0.77 ± 0.02 ^ab^	400.54 ± 11.94 ^d^	425.62 ± 19.09 ^d^	0.999 ± 0.00 ^a^	95.85 ± 9.15 ^d^
Control_6d	264.33 ± 14.17 ^c^	3.31 ± 0.06 ^abcd^	0.80 ± 0.01 ^ab^	309.82 ± 34.28 ^bc^	331.54 ± 45.62 ^bc^	0.999 ± 0.00 ^a^	62.285 ± 7.92 ^c^
High O_2__6d	293.33 ± 4.91 ^c^	3.44 ± 0.07 ^cd^	0.83 ± 0.01 ^ab^	329.53 ± 7.43 ^c^	344.16 ± 8.75 ^c^	0.999 ± 0.00 ^a^	62.52 ± 4.08 ^c^
High O_2__12d	274.33 ± 1.45 ^c^	3.46 ± 0.06 ^d^	0.84 ± 0.01 ^ab^	310.79 ± 11.39 ^bc^	325.59 ± 7.74 ^bc^	0.999 ± 0.00 ^a^	56.05 ± 1.00 ^bc^
Control_18d	193.33 ± 10.87 ^a^	2.86 ± 0.10 ^a^	0.75 ± 0.01 ^a^	216.58 ± 15.52 ^a^	223.06 ± 13.66 ^a^	0.999 ± 0.00 ^a^	38.97 ± 2.30 ^a^
High O_2__18d	287.67 ± 11.31 ^c^	3.39 ± 0.17 ^bcd^	0.81 ± 0.03 ^ab^	321.90 ± 10.61 ^c^	343.61 ± 10.32 ^c^	0.999 ± 0.00 ^a^	64.97 ± 6.28 ^c^
High O_2__24d	272.33 ± 11.92 ^c^	3.27 ± 0.30 ^abcd^	0.77 ± 0.07 ^ab^	303.80 ± 22.32 ^bc^	318.42 ± 22.21 ^bc^	0.999 ± 0.00 ^a^	62.61 ± 5.00 ^c^
Control_30d	171.00 ± 8.00 ^a^	2.97 ± 0.12 ^ab^	0.77 ± 0.04 ^ab^	202.13 ± 21.28 ^a^	213.84 ± 23.79 ^a^	0.999 ± 0.00 ^a^	34.46 ± 1.33 ^a^
High O_2__30d	230.67 ± 10.27 ^b^	3.51 ± 0.18 ^d^	0.85 ± 0.20 ^b^	284.01 ± 25.89 ^bc^	281.36 ± 18.44 ^abc^	0.999 ± 0.00 ^a^	44.81 ± 1.56 ^ab^

Data are means ± SD of three replicates. Different letters represent significant differences among samples.

## Data Availability

Data is contained within the article or Appendix A.

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
