# Peer review of "High Oxygen Shocking Reduces Postharvest Disease and Maintains Satisfying Quality in Fresh Goji Berries during Cold Storage by Affecting Fungi Community Composition"

_foods, 2023, doi:10.3390/foods12132548_

Round 1
Reviewer 1 Report
The manuscript show interesting results to scientific community. Below, I indicate some observations.
1-) Add references to "Non-Metric Multidimensional Scaling", PCoA, LEfSe, and Pearson correlation coefficient (lines 270, 274, 318, and 337).
2-) In Material and methods, describe parameters used in statistical analysis (Non-Metric Multidimensional Scaling", PCoA, LEfSe, and PCC), which are pertinent.
3-) Add throughout the manuscript "sp." or "spp." for name of species in accordance to scientific nomenclature. Example: Alternaria - change to "Alternaria alternata", or Alternaria sp. (when you don't know if it's Alternaria alternata ou Alternaria solani), or Alternaria spp. (when more than one species of Alternaria is present).
Minor corrections
- Format "Lycium barbarum" in italic style (lines 12, 28, and 83);
- Format "Alternaria", "Cladosporium", "Fusarium", and "Fiobasidium" in italic style (lines 36 and 37);
- Change the "M" in "Microbial" to lower case (line 40);
- Change "post-harvest" to "postharvest" (lines 46 and 73). This is in accordance with line 62;
- Format "Botrytis cinerea", "Escherichia coli", "Salmonella enteritidis", "Pseudomonas fluorescens", and "S. enteritidis" in italic style (lines 60 and 61);
- Exclude space in "24 h" and 48 h" (lines 86 and 100);
- Change the "l" in "ml" to capital letter (line 98). This is in accordance with line 122;
- Improve the symbol for "micro" in "microliters" and "micromolar" (lines 124, 132, and 140);
- Change "Multi-Dimensional" to "Multidimensional" (line 270);
- Change "NMDS (Non-Metric Multi-Dimensional Scaling)" to "Non-Metric Multidimensional Scaling (NMDS)" (line 270);
- Exclude underline of "Filobasidium_magnum", "Alternaria_alternata", and "Lysurus_cruciatus" (lines 293 and 321);
- Format "Udenimyces", "Markia", "Cladosporium", "Didymella", "Vishniacozyma", and "Lysurus" to italic style (line 310-113);
- Change "LEfSe" to "linear discriminant analysis effect size (LEfSe)" (line 318);
- Format "Vishniacozyma", "Filobasidium", "Mrakia", "Filobasidium", "Alternaria", "Cladosporium", "Aureobasidium", Penicillium, "Botrytis cinerea", and "Colletotrichum acutatum" in italic style (lines 385-387, 391, 393, 397, and 398).
Author Response
Thank you for your summary. We really appreciate your efforts in reviewing our manuscript (foods-2448709). Those comments are value and very helpful for revising and improving our paper, as well as the important guiding significance to our researches. We have revised the manuscript accordingly. Our point-by-point responses are detailed below.
- Comments and Suggestions for Authors:
- Add references to "Non-Metric Multidimensional Scaling", PCoA, LEfSe, and Pearson correlation coefficient (lines 270, 274, 318, and 337).
Response 1:Thank you for your careful review. We have added relevant references in the "Non-Metric Multidimensional Scaling", PCoA, LEfSe, and Pearson correlation coefficient (lines 270, 274, 318, and 337) according to your comments. Specific as follows:
NMDS is a nonlinear model based on OTUs[1].
Additionally, PCoA, based on the weighted UniFrac distance matrix at the OTU level, was used to assess the sample’s fungal community similarity[2].
LEfSe analysis was performed on microorganisms from the samples[3].
PCC was used to investigate the correlation of dominant genera (genus level, top 10) with the quality properties of goji berries [4].
- In Material and methods, describe parameters used in statistical analysis (Non-Metric Multidimensional Scaling", PCoA, LEfSe, and PCC), which are pertinent.
Response 2:Thank you for your careful review. We have made a more detailed relevant information in the revised manuscript according to your comments. Specific as follows:
And Non-Metric Multidimensional Scaling (NMDS) is a nonlinear model based on Bray-Curtis distance. The Bray-Curtis distance was used to calculate the difference distance between sample [5]. Principal coordinates analysis (PCoA) was used to reduce the dimension of multidimensional microbial data and show the trend of changes through the distribution of samples on the coordinate axis [6]. NMDS and PCoA were plotted using R software (Version 2.15.3). Linear discriminant analysis (LDA) effect size (LEfSe) analysis use LEfSe software. The threshold for feature discrimination was a logarithmic LDA score of 4.0 [3]. Pearson correlation coefficient (PCC) was performed on selected variables of high oxygen shocking and control treatments. The correlation coefficient was visualized in -1 to +1. P-value ≤ 0.05 or P-value ≤ 0.01 were indicated the data was significant [7].
- Add throughout the manuscript "sp." or "spp." for name of species in accordance to scientific nomenclature.Example: Alternaria - change to "Alternaria alternata", or Alternariasp. (when you don't know if it's Alternaria alternata or Alternaria solani), or Alternaria spp. (when more than one species of Alternaria is present).
Response 3:Thank you for your careful review and remind. We have added throughout the manuscript "sp." for name of species at genus level in accordance to scientific nomenclature according to your comments.
- Comments on the Quality of English Language: minor corrections
- Format "Lycium barbarum" in italic style (lines 12, 28, and 83);
- Format "Alternaria", "Cladosporium", "Fusarium", and "Fiobasidium" in italic style (lines 36 and 37);
- Change the "M" in "Microbial" to lower case (line 40);
- Change "post-harvest" to "postharvest" (lines 46 and 73). This is in accordance with line 62;
- Format "Botrytis cinerea", "Escherichia coli", "Salmonella enteritidis", "Pseudomonas fluorescens", and "S. enteritidis" in italic style (lines 60 and 61);
- Exclude space in "24 h" and 48 h" (lines 86 and 100);
- Change the "l" in "ml" to capital letter (line 98);
- Improve the symbol for "micro" in "microliters" and "micromolar" (lines 124, 132, and 140);
- Change "Multi-Dimensional" to "Multidimensional" (line 270);
- Change "NMDS (Non-Metric Multi-Dimensional Scaling)" to "Non-Metric Multidimensional Scaling (NMDS)" (line 270);
- Exclude underline of "Filobasidium_magnum", "Alternaria_alternata", and "Lysurus_cruciatus" (lines 293 and 321);
- Format "Udenimyces", "Markia", "Cladosporium", "Didymella", "Vishniacozyma", and "Lysurus" to italic style (line 310-113);
- Change "LEfSe" to "linear discriminant analysis effect size (LEfSe)" (line 318);
- Format "Vishniacozyma", "Filobasidium", "Mrakia", "Filobasidium", "Alternaria", "Cladosporium", "Aureobasidium", Penicillium, "Botrytis cinerea", and "Colletotrichum acutatum" in italic style (lines 385-387, 391, 393, 397, and 398).
Response:Thank you for your careful review and remind. We have rechecked all errors section according to your comments and made detailed relevant corrections in the revised manuscript.
We really appreciate all your generous comments and suggestions. We wish good health to you and your family. Your careful review has helped to make our study clearer and more comprehensive.
Thank you and best regards.
Sincerely,
Xia Liu, Ph. D.
State Key Laboratory of Food Nutrition and Safety,
Tianjin University of Science and Technology,
Ministry of Education, No. 29, 13th Street, 300457, TEDA, Tianjin,
- R. China
Tel.: +86- 22 60602105
Fax: +86- 22 60601341
Email: liuxia831930@163.com (X. L.)
References
- Noval RM, Burton OT, Wise P, Zhang YQ, Hobson SA, Garcia LM, et al. A microbiota signature associated with experimental food allergy promotes allergic sensitization and anaphylaxis. J Allergy Clin Immunol2013, 131(1): 201-212.
- R MP. An evaluation of the relative robustness of techniques for ecological ordination. Vegetatio1987, 69: 89-107.
- Segata N, Izard J, Waldron L, Gevers D, Miropolsky L, Garrett WS, et al. Metagenomic biomarker discovery and explanation. Genome Biol2011, 12(6): R60.
- Zhang Y, Mahidul IMM, Gao C, Cheng Y, Guan J. Ozone reduces the fruit decay of postharvest winter jujube by altering the microbial community structure on fruit surface. Microbiol Res2022, 262: 127110.
- Bray JR, Curtis JT. An Ordination of the Upland Forest Communities of Southern Wisconsin. Ecol Monogr1957, 27(4): 326-349.
- Ramette A. Multivariate analyses in microbial ecology. Fems Microbiol Ecol2007, 62(2): 142-160.
- Harsonowati W, Masrukhin, Narisawa K. Prospecting the unpredicted potential traits of Cladophialophora chaetospira SK51 to alter photoperiodic flowering in strawberry, a perennial SD plant. Sci Hortic-Amsterdam2022, 295: 110835.

Reviewer 2 Report
This manuscript is about high oxygen shocking reduces postharvest disease and maintains satisfying in fresh goji berries during cold storage by affecting fungal community composition.
It is well presented, however I have some remarks:
Line 28, 83 and in all the text.... add "Lycium barbarum" in italic.
Line 60-61 add all the bacteria cited in italic and in all the text.
Line 31, 47, 83, 97, 132,138 ......Set the space between words and in all the text.
Line 100, add a capital letter after point.
Line 101- 104: the sentence should be reformulated.
The authors should add references to the methods cited: line 113-116, 117-120, 136-134,
Line 137-139: Reference should be added the primer.
Line 136-150: reference should be added to the method.
More references should be added to the discussion.
Perspectives should be added to the conclusion.
Revision and editing the english language of the manuscript is recommended
Author Response
Thank you for your summary. We really appreciate your efforts in reviewing our manuscript (foods-2448709). Those comments are value and very helpful for revising and improving our paper, as well as the important guiding significance to our researches. We have revised the manuscript accordingly. Our point-by-point responses are detailed below.
- Comments and Suggestions for Authors:
This manuscript is about high oxygen shocking reduces postharvest disease and maintains satisfying in fresh goji berries during cold storage by affecting fungal community composition.
It is well presented, however I have some remarks:
- Line 28, 83 and in all the text.... add "Lycium barbarum" in italic.
Response 1:Thank you for your careful review. We have formatted "Lycium barbarum" in italic style in the revised manuscript according to your comments.
- Line 60-61 add all the bacteria cited in italic and in all the text.
Response 2:Thank you for your careful review. We have formatted all bacteria, such as "Botrytis cinerea", "Escherichia coli", "Salmonella enteritidis", "Pseudomonas fluorescens", and "S. enteritidis" in italic (checked all the text) in the revised manuscript according to your comments. Specific as follows:
-"Botrytis cinerea", "Escherichia coli", "Salmonella enteritidis", "Pseudomonas fluorescens", and "S. enteritidis" (lines 60 and 61);
-"Udenimyces", "Markia", "Cladosporium", "Didymella", "Vishniacozyma", and "Lysurus" (line 110-113);
-"Vishniacozyma", "Filobasidium", "Mrakia", "Filobasidium", "Alternaria", "Cladosporium", "Aureobasidium", Penicillium, "Botrytis cinerea", and "Colletotrichum acutatum" (lines 385-387, 391, 393, 397, and 398).
- Line 31, 47, 83, 97, 132,138 ......Set the space between words and in all the text.
Response 3:Thank you for your careful review. We have checked and corrected the space in all the text according to your comments.
- Line 100, add a capital letter after point.
Response 4:Thank you for your careful review. We have added a capital letter after point in the revised manuscript according to your comments. Specific as follows: “After incubation.....”
- Line 101-104: the sentence should be reformulated.
Response 5:Thank you for your careful review. We have reformulated the sentence (Line 101-104) in the revised manuscript according to your comments. Specific as follows:
Calculate the amount as the weighted mean from the successive dilutions containing between 30 and 300 colony-forming units (CFU). The plate below 30 CFU records the specific number of colonies, and the plate greater than 300 CFU can not be recorded. The results were expressed as a number between 1.0 and 9.9 times the appropriate power of 10. The aerobic plate count was evaluated via equation (1).
N = (1)
Where, N — the CFU of samples; — the total CFU of plates; N1 — the number of low dilution plates; N2 — the number of high dilution plates; d — dilution factor (low dilution plates).
- The authors should add references to the methods cited: line 113-116, 117-120, 130-134.
Response 6:Thank you for your careful review. We have added relevant references in the methodology section in the revised manuscript according to your comments. Specific as follows:
line 113-116 — The firmness of fresh goji berries was determined by the GY-4 fruit durometer (Toppu Instrument Co, Ltd. Zhejiang, China). A cylindrical aluminum probe (P/11) with a diameter of 11 mm was used to measure the firmness of the fruit[1]. The firmness was expressed by the peak force, denoted as Newtons (N).
line 117-120 — The TSS was measured according to[2]. After the fruit was juiced through a manual juicer, it was placed on the Pocket refractometer (PAL-1, ATAGO, Japan) to read and record the TSS value.
line 130-134 — The total fungal communities of goji berries determined according to [3], which were collected at 0, 6, 12, 18, 24, and 30 d storage at 0 ± 0.5 ℃. The communities were obtained by using a filter funnel with a microporous membrane (diameter 50 mm, pore diameter 0.45 μm) to filter the the washing water of the goji berries. Vacuum filtration was used to concentrate microbial communities onto the microporous membrane, which was then stored at - 80 ℃.
- Line 137-139: Reference should be added the primer.
Response 7:Thank you for your careful review. We have added relevant references in the methodology section (Line 137-139) in the revised manuscript according to your comments. Specific as follows:
ITS genes of distinct regions were amplified used specific primer (ITS - 1737 F: 5’ - GGAAGTAAAAGTCGTAACAAGG - 3’, ITS - 2043 R: 5’ - GCTGCGTTCTTCATCGATGC - 3’) with the barcode [3].
- Line 136-150: reference should be added to the method.
Response 8:Thank you for your careful review. We are sorry for the ambiguity in this method. In fact, the reference you mentioned is line with the first sentence in this paragraph — “Total genome DNA from samples was extracted using CTAB (Cetyl-trimethyl ammonium bromide) method [27]”.
- More references should be added to the discussion.
Response 9:Thank you for your careful review. We have added relevant references in the discussion section in the revised manuscript according to your comments. Specific as follows:
line 357-358 —In recent years, it has become possible to directly examine the fruits microbiome to reveal how postharvest treatment reduced disease and decay [4].
line 367-371 —Here, high oxygen shocking improving the good quality properties of fresh goji berries, such as firmness, weight loss, and TA. Some studies have reported that cell wall is the first barrier of pathogen infection. While, it would be solubilized by the cell wall degrading enzymes during storage, and lead to the firmness loss, and weaken the disease resistance, increase the decay incidence of fruits [5-6]. Thus, higher firmness can be help to delay fruit softening and improve fruit resistance. Moreover, modification of storage atmospheres has the ability to induce plant defensive responses and enhance disease resistance in postharvest commodities [36], which support our the second model.
- Perspectives should be added to the conclusion.
Response 10:Thank you for your careful review. We agree with your comment. We have added the perspectives in the conclusion in the revised manuscript according to your comments. Specific as follows:
Our research first explored high oxygen shocking on reducing the occurrence of disease and decay and maintaining the well quality properties of goji berries during cold storage by affecting auto-resistance and fungi community composition. In general, high oxygen shocking effectively reduced decay, maintained the weight loss and firmness, and delayed the maturity of goji berries during storage. Additionally, it caused shifts in the microbial diversity and composition of goji berries’ epidermis. Filobasidium sp., Alternaria sp., Lysurus sp., Aureobasidium sp., Udeniomyces sp., Mrakia sp., Cladosporium sp., Didymella sp., and Vishniacozyma sp. were the main genera widespread in most goji berry samples. Concurrently, high oxygen shocking inhibited the pathogenic fungi and harmful saprotrophic fungi by promoting the growth of antagonist of postharvest pathogens. The changes in fungi community succession inhibited of the development of pathogenesis of goji berries. Based on these results, this study provides insights into the mechanism by which high oxygen shocking controls postharvest losses caused by fungi in fresh fruit.
Comments on the Quality of English Language: Revision and editing the english language of the manuscript is recommended
Response:Thank you for your careful review. We have improved the English writting and some grammatical errors in the paper have been fixed by my professional tutor in the revised manuscript.
References
- Goldberg T, Agra HE, Ben-Arie R. Non-destructive measurement of fruit firmness to predict the shelf-life of ‘Hayward’kiwifruit. Sci Hortic-Amsterdam2019, 244: 339-342.
- Barros R, Andrade J, Denadai M, Nunes ML, Narain N. Evaluation of bioactive compounds potential and antioxidant activity in some Brazilian exotic fruit residues. Food Res Int2017, 102: 84-92.
- Liu X, Gao Y, Yang H, Li L, Jiang Y, Li Y, et al. Pichia kudriavzeviiretards fungal decay by influencing the fungal community succession during cherry tomato fruit storage. Food Microbiol2020, 88: 103404.
- 4. Dai K, Han P, Zou X, Jiang S, Xu F, Wang H, et al. Hot air treatment reduces postharvest decay in Chinese bayberries during storage by affecting fungal community composition. Food Res Int2021, 140: 110021.
- 5. Carpita NC, Gibeaut DM. Structural models of primary cell walls in flowering plants: consistency of molecular structure with the physical properties of the walls during growth. Plant J1993, 3(1): 1-30.
- 6. W. Zhou LSRB. Analysis of cell wall components in juice of flavortop nectarines during normal ripening and woolliness. J Amer Soc Hort Sci1999, 124(4): 424-429.

Reviewer 3 Report
Authors studied the potential effects of high oxygen shocking on the quality of chill-stored fresh goji berries. Results indicated that high-concentrated oxygen shocking (90%) led to a more effective quality maintenance, affecting in parallel the mycobiota by reducing the abundance of several undesired/harmful fungi genera. In general, the work is interesting, the aim is clear and the experimental design is appropriate. However, there are some issues that have to be carefully addressed by authors, which are mainly related to a limited discussion of some of the findings, making the scientific soundness a bit weak. Furthermore, a moderate English editing is recommended.
-L25. The first and last keywords should be revised.
-L32. Please revise.
-L35. Which is market demand? Please specify and add some ref(s).
-L36-37. Genus with italics. Please check carefully throughout the manuscript.
-L40. From my point of view, the term “contamination” should be preferred rather than “infection”. Please check throughout the manuscript.
-L66. “biological succession” please revise.
-L79. This a microbiota profile study rather than a microbiome one.
-L95-104. Why authors studied just APC? What about other microbial groups, e.g., yeasts, fungi, or even bacteria? Please specify.
-L103-104. Not clear, please revise.
-L137. ITS region.
-L143-144. “After incubation (72 ℃, 5 min)” please revise. It is the final elogation.
-L172. DMRTor LSD test? Please check and revise.
-L290. In Figure 5 a,b the results should be presented as % of relative abundances (0-100%).
-Λ345. What is the meaning of unidentified? If there are several unidentified OTUs, it is better to be removed from the analysis.
-L367-370. A thorough discussion is needed herein.
-L378-379. Similarly, a possible explanation(s) is strongly recommended herein.
-L382-384. Why oxygen shocking altered the mycobiota composition of goji during storage? Please discuss more deeply.
-L398-401. Which is the mechanism responsible for pathogens’ inhibition by using oxygen shocking?
-L401. A final paragraph highlighting a) the importance of this work, b) the main findings, c) how scientific/industry communities could be benefited by the findings of this study and d) which should be the “next steps”, is strongly recommended.
Moderate English editing is recommended.
Author Response
Thank you for your summary. We really appreciate your efforts in reviewing our manuscript (foods-2448709). Those comments are value and very helpful for revising and improving our paper, as well as the important guiding significance to our researches. We have revised the manuscript accordingly. Our point-by-point responses are detailed below.
- Comments and Suggestions for Authors:
Authors studied the potential effects of high oxygen shocking on the quality of chill-stored fresh goji berries. Results indicated that high-concentrated oxygen shocking (90%) led to a more effective quality maintenance, affecting in parallel the mycobiota by reducing the abundance of several undesired/harmful fungi genera. In general, the work is interesting, the aim is clear and the experimental design is appropriate. However, there are some issues that have to be carefully addressed by authors, which are mainly related to a limited discussion of some of the findings, making the scientific soundness a bit weak. Furthermore, a moderate English editing is recommended.
- -L25. The first and last keywords should be revised.
Response 1:Thank you for your careful review. We have made any adjustments in the revised manuscript according to your comment about keywords. Specific as follows:
Keywords: Fungal community; high oxygen shocking; goji berry; quality
- -L32. Please revise.
Response 2:Thank you for your careful review. We have revised the quotes according your comments. Specific as follows:
“super fruit” instead of ‘super fruit’
- -L35. Which is market demand? Please specify and add some ref(s).
Response 3:Thank you for your careful review. Market demand means a large number of people can not eat the fresh goji berries because of the short shelf life. Fresh goji berries are highly perishable, they can only be sold near where they are cultivated, affecting their market potential. We have added relevant a reference in the revised manuscript according to your comments. Specific as follows:
However, fresh goji berries are vulnerable to fungal infection, leading to quality deterioration and short shelf life, which makes it unable to meet the market demand [1].
- -L36-37. Genus with italics. Please check carefully throughout the manuscript.
Response 4:Thank you for your careful review and remind. We have formatted all bacteria, such as “Alternaria sp., Cladosporium sp., Fusarium sp., and Fiobasidium sp.” in italic (checked all the text) in the revised manuscript according to your comments. Specific as follows:
-“Alternaria sp., Cladosporium sp., Fusarium sp., and Fiobasidium sp.” (L36-37)
-"Botrytis cinerea", "Escherichia coli", "Salmonella enteritidis", "Pseudomonas fluorescens", and "S. enteritidis" (L60 and 61);
-"Udenimyces sp.", "Markia sp.", "Cladosporium sp.", "Didymella sp.", "Vishniacozyma sp.", and "Lysurus sp." (L110-113);
-"Vishniacozyma sp.", "Filobasidium sp.", "Mrakia sp.", "Filobasidium sp.", "Alternaria sp.", "Cladosporium sp.", "Aureobasidium sp.", Penicillium sp., "Botrytis cinerea", and "Colletotrichum acutatum" (L385-387, 391, 393, 397, and 398).
- -L40. From my point of view, the term “contamination” should be preferred rather than “infection”. Please check throughout the manuscript.
Response 5:Thank you for your careful review. We agree with your comment. We have used “contamination” instead of “infection” in the revised manuscript.
- -L66. “biological succession” please revise.
Response 6:Thank you for your careful review. We have used “fungal community” instead of “biological succession” in the revised manuscript.
- -L79. This a microbiota profile study rather than a microbiome one.
Response 7:Thank you for your careful review. We agree with your comment. We have used “microbiota” instead of “microbiome” in the revised manuscript.
- -L95-104. Why authors studied just APC? What about other microbial groups, e.g., yeasts, fungi, or even bacteria? Please specify.
Response 8:Thank you for your careful review. APC in the text was conducted following the GB 4789.2 standard operation procedure of China. It is the total number of colonies of all microorganisms, including fungi, yeasts, and bacteria. It has two aspects of hygienic significance: on the one hand is the hygiene of food, the more the quantity, the worse the condition, on the other hand is the shelf life of food. We tested it in order to predict shelf life of fruits.
Postharvest fungal contamination is one of important factors causes substantial losses of fruits and vegetables during storage. Thus, we focused on the study of fungi, yeasts and bacteria were not studied in this text.
- -L103-104. Not clear, please revise.
Response 9:Thank you for your careful review. We are sorry for the ambiguity in this sentence. We have made any adjustments in the revised manuscript according to your comment. Specific as follows:
Calculate the amount as the weighted mean from the successive dilutions containing between 30 and 300 colony-forming units (CFU). The plate below 30 CFU records the specific number of colonies, and the plate greater than 300 CFU can not be recorded. The results were expressed as the negative logarithm of the CFU of samples. The aerobic plate count was evaluated via equation (1).
N = (1)
Where, N—the CFU of samples; —the total CFU of plates; N1—the number of low dilution plates; N2—the number of high dilution plates; d—dilution factor (low dilution plates).
- -L137. ITS region.
Response 10:Thank you for your careful review. We identified the ITS region in the revised manuscript according to your comment. Specific as follows:
ITS1 gens of distinct regions were amplified used specific primer.
- -L143-144. “After incubation (72℃, 5 min)” please revise. It is the final elogation.
Response 11:Thank you for your careful review. We have made any adjustments in the revised manuscript according to your comment about the sentence. Specific as follows:
“Finally 72 ℃ for 5 min”.
- -L172. DMRTor LSD test? Please check and revise.
Response 12:Thank you for your careful review and remind. We apologized for the writting mistakes in the manuscript. We have made corrected in the revised manuscript. Specific as follows:
SPSS 22.0 was used for Duncan’s multiple range tests to compare the mean separations.
- -L290. In Figure 5 a,b the results should be presented as % of relative abundances (0-100%).
Response 13:Thank you for your careful review. We have made corrected the Figure 5 in the revised manuscript. Specific as follows:
- -L What is the meaning of unidentified? If there are several unidentified OTUs, it is better to be removed from the analysis.
Response 14:Thank you for your careful review. We agree with your comment. We have removed the relevant information from the analysis.
- -L367-370. A thorough discussion is needed herein.
Response 15:Thank you for your careful review. We have rewritten this paragraph and made a more detailed relevant information in the revised manuscript according to your comments. Specific as follows:
Here, high oxygen shocking improving the good quality properties of fresh goji berries, such as firmness, weight loss, and TA. Some studies have reported that cell wall is the first barrier of pathogen infection. While, it would be solubilized by the cell wall degrading enzymes during storage, and lead to the firmness loss, and weaken the disease resistance, increase the decay incidence of fruits [2-3]. Thus, higher firmness can be help to delay fruit softening and improve fruit resistance. Moreover, modification of storage atmospheres has the ability to induce plant defensive responses and enhance disease resistance in postharvest commodities [47], which support the second model.
- -L378-379. Similarly, a possible explanation(s) is strongly recommended herein.
Response 16:Thank you for your careful review. We have made a more detailed relevant information in the revised manuscript according to your comments. Specific as follows:
Meanwhile, previous reports have indicated a correlation between microbiome diversity and plant health. Diseased plants tend to exhibit a lower diversity of microbes compared to healthy plants, which is associated with the activation of plant immune signals [38].
- -L382-384. Why oxygen shocking altered the mycobiota composition of goji during storage? Please discuss more deeply.
Response 17:Thank you for your careful review. The comment you made is very worth thinking about. In this text, as the three principal mechanisms models of high oxygen shocking effectively inhibited the aerobic plate count and decay was supposed (L359-366), first model was 90% O2 shocking (30min) suppressed the growth of low-oxygen consumption bacteria or anaerobic bacteria; second model was 90% O2 shocking (30min) enhanced the resistance of fruits (data on disease resistance and oxidation resistance were not published), hindered the contamination of pathogens, inhibited the pathogens absorption of fruit nutrients, and thus reducing the aerobic plate count and decay; third model was disturbed the changes of microbial ecological community species structure during storage, and inhibited the growth of pathogenic fungi. Presumably, the first and second model may the reason why oxygen shocking altered the mycobiota composition to ensure fruits health needs to be further explored.
- -L398-401. Which is the mechanism responsible for pathogens’ inhibition by using oxygen shocking?
Response 18:Thank you for your careful review. Besides, the three principal mechanisms models of high oxygen shocking effectively inhibited the aerobic plate count and decay was supposed (L359-366), Aureobasidium sp. was significantly enriched in fruit at early storage by high oxygen shocking, which as an effective antagonist of postharvest pathogens of fruits, contributed against phytopathogenic fungi and had been used to control postharvest diseases in citrus, tomatoes, and apples. Presumably, it antagonized fruit postharvest pathogens, such as Penicillium spp., Botrytis cinerea, and Colletotrichum acutatum. Consequently, high oxygen shocking may inhibit pathogens by promoting the growth of antagonist of postharvest pathogens. The changes in fungal community succession influenced the inhibition of the development of pathogenesis of goji berries.
- -L401. A final paragraph highlighting a) the importance of this work, b) the main findings, c) how scientific/industry communities could be benefited by the findings of this study and d) which should be the “next steps”, is strongly recommended.
Response 19:Thank you for your careful review. We have made a more detailed relevant information in the revised manuscript according to your comments. Specific as follows:
These results confirmed that high oxygen shocking has significant potential for maintaining postharvest quality of fresh goji berries, which was an effective, safe, low-cost, convenient and environmentally friendly method for berries preservation. Our findings hold great promise for the advancement of postharvest fruit preservation technology. Besides, unidentified fungi genera (Table S1) should not be overlooked, and further studies are required to gain a comprehensive understanding of their role.
- Comments on the Quality of English Language: moderate English editing is recommended.
Response:Thank you for your careful review. We have improved the English writting and some grammatical errors in the paper have been fixed by my professional tutor in the revised manuscript.
References
- Zhang H, Ma Z, Wang J, Wang P, Lu D, Deng S, et al. Treatment with exogenous salicylic acid maintains quality, increases bioactive compounds, and enhances the antioxidant capacity of fresh goji (Lycium barbarum L.) fruit during storage. Food science & technology2021, 140: 110837.
- Carpita NC, Gibeaut DM. Structural models of primary cell walls in flowering plants: consistency of molecular structure with the physical properties of the walls during growth. Plant J1993, 3(1): 1-30.
- H. W. Zhou LSRB. Analysis of cell wall components in juice of flavortop nectarines during normal ripening and woolliness. J Amer Soc Hort Sci1999, 124(4): 424-429.

Round 2
Reviewer 2 Report
The authors have improved the manuscript and made it really clear
Reviewer 3 Report
The revised manuscript has been strongly improved and authors addressed all of my remarks. I have no further comments.
No comments